# Impacts of COVID-19-related service disruptions on TB incidence and deaths in Indonesia, Kyrgyzstan, Malawi, Mozambique, and Peru: Implications for national TB responses

Rowan Martin-Hughes[1]*, Lung Vu[2]*, Nejma Cheikh[2], Sherrie L. Kelly[1], Nicole Fraser-Hurt[2], Zara Shubber[2], Ivan Manhiça[3], Kuzani Mbendera[4], Belaineh Girma[4], Imran Pambudi[5], Julia Ríos[6], Abdrahmanova Elmira[7], Pandu Harimurti[2], Reem Hafez[2], Jaime Nicolas Bayona Garcia[2], Tom Palmer[8], Anna Roberts[1], Marelize Gorgens[2], David Wilson[2]

1 Burnet Institute, Melbourne, Victoria, Australia, 2 The World Bank, N.W. Washington, DC, United States of America, 3 NTP Mozambique, Maputo, Mozambique, 4 NTP Malawi, Ministry of Health, Lilongwe, Malawi, 5 NTP Indonesia, Ministry of Health, Jakarta Selatan, Indonesia, 6 NTP Peru, Ministry of Health, Jesús María, Lima, Peru, 7 NTP Kyrgyzstan, Bishkek, Kyrgyzstan, 8 Institute for Global Health, University College London, London, United Kingdom

* rowan.martin-hughes@burnet.edu.au (RM-H); lvu8@worldbank.org (LV)

## Abstract

Initial global-level estimates reported in June 2020 by the World Health Organization suggested that levels of disruption to TB service delivery could be as high as 25%-50% and result in an additional 6·3 million cases of tuberculosis (TB) and an additional 1·4 million TB-related deaths attributable to COVID-19 between 2020 and 2025. Quarterly epidemiological estimates and programmatic TB data capturing disruption levels to each TB service were collected by National TB Programmes in Indonesia, Kyrgyzstan, Malawi, Mozambique, and Peru. Data from 2019, for a pre-COVID-19 baseline, and throughout 2020, together with the NTP's COVID-19 response plans, were used within Optima TB models to project TB incidence and deaths over five years because of COVID-19-related disruptions, and the extent to which those impacts may be mitigated through proposed catch-up strategies in each country. Countries reported disruptions of up to 64% to demand-driven TB diagnosis. However, TB service availability disruptions were shorter and less severe, with TB treatment experiencing levels of disruption of up to 21%. We predicted that under the worse-case scenario cumulative new latent TB infections, new active TB infections, and TB-related deaths could increase by up to 23%, 11%, and 20%, respectively, by 2024. However, three of the five countries were on track to mitigate these increases to 3% or less by maintaining TB services in 2021 and 2022 and by implementing proposed catch-up strategies. Indonesia was already experiencing the worse-case scenario, which could lead to 270,000 additional active TB infections and 36,000 additional TB-related deaths by the end of 2024. The COVID-19 pandemic is projected to negatively affect progress towards 2035 End TB targets, especially in countries already off-track. Findings highlight both successful TB service delivery

**Data Availability Statement:** The Optima TB interface is available via http://ocds.co/tb. The Optima TB epidemiological model is developed within the Atomica framework (https://github.com/atomicateam/atomica). The user interface and webserver components are written using Sciris (https://github.com/sciris/sciris). The tool is available as part of the "Atomica Applications" suite (https://github.com/sciris/atomica_apps). Data and reports for each underlying country model are published and openly available in the World Bank Open Knowledge Repository for each country as cited, and full model inputs for each scenario based on country data are given in Appendix S1. Proposals to access original country data owned by National Tuberculosis Programmes should be directed to Lung Vu (lvu8@worldbank.org) and will require a formal data access agreement with the respective National Tuberculosis Programmes.

**Funding:** This work was funded through UNAIDS grant number AA-P170789-GPLG-TF0B2204. The funders had no role in study design, data collection and analysis, decision to publish, or preparation of the manuscript.

**Competing interests:** The authors have declared that no competing interests exist.

adaptions in 2020 and the need to proactively maintain TB service availability despite potential future waves of more transmissible COVID-19 variants.

## Introduction

Tuberculosis (TB) is a communicable disease and one of the top 10 causes of death worldwide [1]. In 2019 alone, about 10 million people developed TB and 1·4 million died from TB [1,2]. With the serious consequences of TB on human health and development outcomes, the World Health Organization (WHO) and UN Member States are committed to ending the TB epidemic through their adoption of WHO's End TB Strategy (2015–2035), as well as the UN Sustainable Development Goals (SDGs) that were established in 2014 [1]. Although much progress has been made during the last decade in reducing TB incidence and deaths and while a number of high TB burden countries were on track to reach the 2020 TB milestones, the COVID-19 pandemic could threaten to reverse progress made and will likely have a short- and medium-term impact (through to 2025) on new TB cases and TB-related deaths [1,3]. The WHO reports that global TB notifications fell by 18% in 2020 relative to 2019, with ongoing shortfalls in 2021 [4]. The COVID-19 pandemic will continue to affect the TB epidemic in several ways, particularly through disruptions of health services as the result of ongoing COVID-19 mitigation strategies and shifting of heath resources in response to COVID-19. These include closure of public transport, closure of health facilities, reduced clinic hours, reduced care-seeking behaviours and service uptake due to restrictions and fear of COVID-19 transmission, and interruptions in the supply chain. In addition, health resources have been reallocated or prioritized for the COVID-19 response and increased demand from COVID-19 patients will continue to reduce the capacity of health systems to deliver prevention and treatment services for other diseases, including TB. Furthermore, severe economic consequences and loss of income are likely to worsen some of the factors that determine TB susceptibility, including malnutrition and poverty.

During the early phase of the COVID-19 pandemic in early 2020, findings from the Stop TB partnership predicted a global decrease in the TB case detection rate of about 25% from April to June 2020, compared with detection levels from the pre-COVID equivalent months in 2019. This analysis also predicted an additional 6·3 million cases of TB and an additional 1·4 million TB-related deaths attributable to COVID-19 between 2020 and 2025 [5,6]. In recent months, several studies have reported disruption levels of TB services and have commented on the potential impact these disruptions may have on TB incidence and death in the short- and medium-term. For example, between April and June 2020, India reported a more than 50% decrease in people with TB as being registered on treatment, and a 30% decrease in TB treatment completion compared with the same period in 2019 [6,7]. It was estimated that in 2020, combined with existing poverty exacerbation of food insecurity in India, and disrupted TB services due to the COVID-19 pandemic could result in a 14% increase in TB incidence and a 20% increase in TB deaths [8].

Compared with the same 3-month period in 2019, overall reductions of TB case notifications in Indonesia and the Philippines ranged from 25% to 30% [6]. In Nigeria, analysis using data from 61 high-volume facilities across nine states showed a decrease of 63%, 64%, 73%, and 72% in clinic attendance, presumptive TB identification, confirmed TB cases, and treatment initiation, respectively [9]. Several other studies conducted in Northeastern Brazil [10], Sierra Leone [11], Kenya [5], Ukraine [6], and China [12], at either the subnational or national

level, also showed reductions between 30% and 50% in TB service uptake due to physical distancing measures. Across 43 TB centers in 19 countries from five continents, TB diagnoses were reduced in all centers except for in two, in Australia and Virginia, USA [13]. Based on data for 80 countries collected by the WHO, an estimated 1·4 million fewer people (21% less) received TB care in 2020 compared with 2019 [14], and Indonesia and Peru had the second and thirteenth biggest shortfall in TB notifications worldwide in 2020, respectively, compared with 2019 [4]. Furthermore, TB referrals for people with suspected TB decreased by 59% [14]. This means much of the progress that had been made in identifying TB cases in the past decade was reversed in 2020.

Modelling studies conducted at the global or national level early in the pandemic were largely limited to projecting the impact the first three-months of lockdown measures would have on public health. These first analyses were a call-to-action based on assumptions of a worse-case scenario where no mitigation strategies were put in place by NTPs, but even the worst-case anticipated length of disruption was much shorter than what has actually been experienced, often on the order of 3–6 months [2,5,6,15]. Throughout 2020 and 2021 the TB Modelling and Analysis Consortium collated available evidence on disruption to TB services experienced due to COVD-19 and assessed ongoing global TB data gaps in real-time to inform efficient allocation of existing resources and rapid deployment of additional funds necessary to best mitigate the impact of disruption [16].The aim of this study is to overcome these limitations, and to be more responsive to the National TB Programmes (NTPs) for five countries–Indonesia, Kyrgyzstan, Malawi, Mozambique, and Peru, and also to examine the impact on both essential TB services and TB outcomes of COVID-19 mitigation actions within country responses to the Global Fund's catch-up plan [17]. As the COVID-19 pandemic is far from being over, our findings will be vital to governments in better understanding the extent to which TB service disruption could lead to additional TB cases and deaths, and how rapid action in program adaptation and scale-up of TB services can help mitigate these negative impacts.

## Methods

### Study design

This modeling study aimed to measure the impact COVID-19-related TB service disruptions on TB incidence and deaths, and to better understand how effective the national TB responses have been in countering the negative impact of COVID-19 on TB in five countries: Indonesia, Kyrgyzstan, Malawi, Mozambique, and Peru. These five countries were selected due to data readiness through previous collaborations and because they present a variety of different TB and COVID-19 epidemics. Disruptions to each of the following TB services and stages of the TB diagnosis and treatment cascade were considered as part of this analysis: drug susceptible (DS)-TB diagnosis rates, drug resistant (DR)-TB diagnosis rates, DS-TB treatment initiation rates, DR-TB treatment initiation rates, DS-TB treatment completion rates, DR-TB treatment completion rates, BCG vaccination numbers, ART coverage proportion among people living with HIV, and the number of people initiated on TB preventive therapy.

Optima TB (http://ocds.co/tb/), a mathematical model of TB transmission and disease progression integrated with an economic and program analysis framework [18], was used to conduct these five country analyses. All analyses were led by the respective NTPs, with technical assistance in running the model, in quality assurance, and in interpreting results from the World Bank and the Burnet Institute. National level Optima models developed for country-led allocative efficiency analyses between 2016 and 2020 were used for Indonesia [19], Kyrgyzstan [20], Mozambique [21], and Peru [22], and district-level models were used for three districts in Malawi (Blantyre, Lilongwe, and Mzimba) [23].

## Model inputs

**Estimates of COVID-19-related service disruptions.**   Two to three discussions were held between each of the five respective NTPs and the modelling team between June 2020 and June 2021 to source necessary COVID-19-related service disruption inputs for this modelling study and to gain qualitative insight into the impact COVID-19 has had and is having on TB service demand and delivery in each country. NTPs shared quarterly data that covered the historical pre-COVID-19 baseline and emerging impact of the COVID-19 pandemic from January 2019 to December 2020, as well as preliminary data for January to June 2021 where available, descriptions of how service delivery had changed in the country, and response plans to implement new or scaled-up interventions. Finally, NTPs shared expectations of how TB services may be impacted in the future in the event of a worse-case wave of COVID-19 relative to what they had previously experienced.

Context-specific numbers of COVID-19 cases and reduced population mobility due to COVID-19-related behavior changes were sourced on 9 April 2021 from the Institute for Health Metrics and Evaluation (IHME) COVID-19 Projections tool [24] (Fig 1). National teams provided quarterly TB program data, including number of outpatient visits, number of treatment initiations, treatment outcomes, as well as qualitative estimates of the level of COVID-19-related TB service disruption to TB diagnosis and care outreach programs including active case finding and Bacillus Calmette-Guérin (BCG) vaccination.

Key phases of COVID-19-related disruption to mobility and health system capacity and service access were identified in each setting based on 3-month model time steps (Table 1). Uncertainty intervals around the actual level of disruption were determined for each country based on programmatic data from different phases from 1 April 2020 to 31 December 2020. These data were used in conjunction with trends of COVID-19 infections to inform the context of disruptions to TB services and to estimate the level of disruption from 1 January 2021 through to 30 June 2021 where routine TB data was not yet available. Modelled levels of disruption after 1 July 2021 were based on best-case and worse-case estimates from NTPs of the severity and duration of disruptions on the national TB epidemic given prospective scenarios of mild or severe health system pressures due to COVID-19, rather than based on relative likelihood. Further qualitative evaluation of progress within this uncertainty interval was conducted by NTPs following these modelling analyses.

**Epidemiological inputs.**   Data used to inform the five country-setting models were obtained from the respective NTP and included, where available, national or district-level quarterly data for 2018, 2019, and 2020 for outpatient TB screening, TB case notifications and treatment initiations disaggregated by drug-resistant status (drug sensitive (DS)-TB, multidrug resistant (MDR)-TB, extensively drug-resistant (XDR)-TB), age bands (under 5, 5–14, 15 and over), HIV status, the number of children under 5-years of age, and people living with HIV (PLHIV) who initiated TB preventive therapy.

Data for all settings captured both the reduced ability to provide TB services due to closures following COVID-19-related policy implementations or capacity limitations and the reduction in demand due to stay-at-home orders, health service directives to minimize "non-urgent" visits to clinics, or reluctance to access health services for fear of COVID-19 transmission. Estimates of the capacity to scale-up TB intervention coverage over a 12- to 24-month recovery period since the onset of the COVID-19 pandemic given available financial resources were generated in consultation with NTPs. Without clear data to the contrary, we assumed that there were no underlying changes to TB transmission, as described in the limitations section.

**Scenarios modelled.**   Five scenarios were modelled for each country-setting as described in Table 1. The best- and worse-case scenarios are designed to capture (1) narrow uncertainty

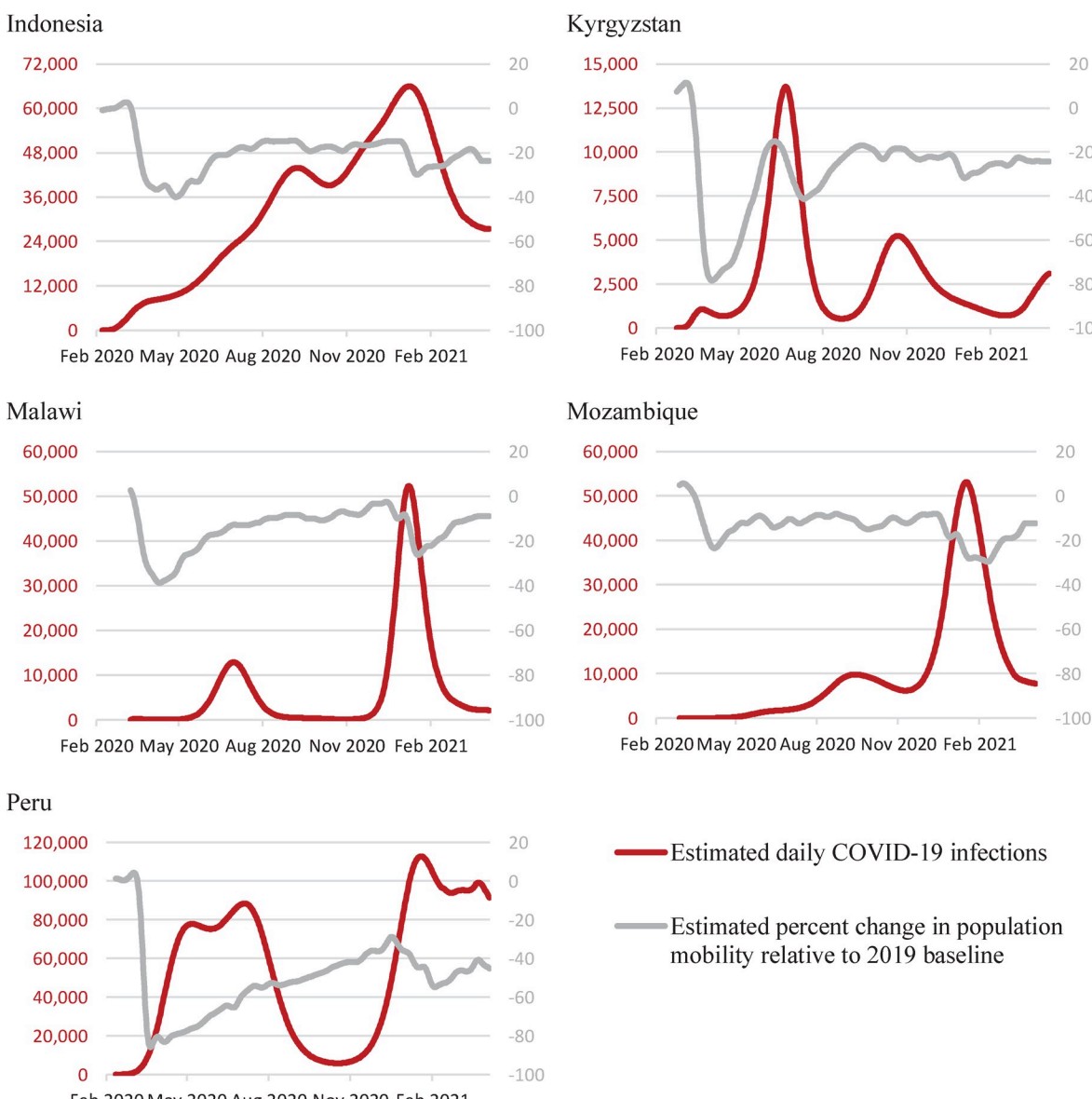

**Fig 1. Estimated daily COVID-19 infections for the study period reported by IHME based on reported COVID-19 daily cases, tests, and Google mobility data from 9 April 2021.**

intervals based on the actual level of disruption experienced in 2020, and (2) wider uncertainty intervals for the levels of COVID-19 transmission that may be experienced in each setting throughout 2021 and 2022 and government responses. These two scenarios were also modelled with the NTP-proposed expanded programs or novel interventions to catch-up to the pre-COVID-19 status quo trend in each country.

## Results

Summaries of qualitative feedback from NTPs and key quantitative differences in reported coverage of TB services for the five countries considered in this study are presented in Table 2. Full translation of coverage data into model inputs are described in the S1 Appendix.

**Table 1. Summary of COVID-19-related disruptions to TB services scenarios modelled.**

| Scenario | 1 January 2019 to 31 March 2020 | 1 April 2020 to 30 June 2021 | 1 July 2021 to 30 June 2022 | 1 July 2022 to 31 December 2022 | 1 January 2023 and ongoing |
|---|---|---|---|---|---|
| **Status quo counterfactual** | Reported notification, treatment, vaccination, and preventive therapy data | 2019 TB service coverage levels [a] | | | |
| **Best-case** | As for status quo | Lower bound of the uncertainty interval for the level of disruption to each TB service, extrapolated based on the estimated COVID-19 infections in Q1 and Q2 2021 and the levels of TB disruption experienced in 2020. | Estimated level of ongoing disruption to TB services with no substantial new waves of COVID-19 during this time, reflecting both vaccination and effective control of transmission through other public health measures. This includes a return to status quo treatment, vaccination, and preventive therapy, but some ongoing reduction in TB diagnosis rates except in Indonesia where the NTP expects continued disruption to all TB services even under a best-case scenario. | 2019 TB service coverage levels [a] | 2019 TB service coverage levels [a] |
| **Best-case with catch-up** | As for status quo | As for best-case | Underlying disruption as for best-case, with the addition of catch-up interventions as detailed in Table 2 | | 2019 TB service coverage levels [a] |
| **Worse-case** | As for status quo | Upper bound of the uncertainty interval for the level of disruption to each TB service, extrapolated based on the estimated COVID-19 infections in Q1 and Q2 2021 and the levels of TB disruption experienced in 2020. | Estimated level of ongoing disruption to TB services at the greater of 10% or the level of disruption experienced throughout 2020, based on the possibility of ongoing waves of COVID-19 transmission as experienced in 2020, such as through new variants including Delta. | 2019 TB service coverage levels [a] | |
| **Worse-case with catch-up** | As for status quo | As for worse-case | Underlying disruption as for worse-case, with the addition of catch-up interventions as detailed in Table 2 | | 2019 TB service coverage levels [a] |

[a] In Malawi a planned expansion of TB preventive therapy in 2020 was included as part of the ongoing status quo in all scenarios from 1 April 2020. In Mozambique a planned expansion of TB outreach screening services in 2020 was included as part of the ongoing status quo in all scenarios from 1 July 2020. In Peru a planned reduction of social determinants of TB transmission was included as part of the ongoing status over 24 months from 1 January 2023 to 31 December 2024 then held constant. In the catch-up scenarios for Peru, this reduction in TB transmission occurred 18-months earlier, from 1 July 2021 until 30 June 2023.

### Reported availability and adaptation of TB services and the TB care cascade due to COVID-19

The most heavily impacted aspect of TB service delivery was passive case finding, e.g., demand-driven TB diagnosis in clinics and hospitals. Passive diagnoses in the second quarter of 2020 experienced disruptions ranging from 11% lower in Mozambique (under the circumstance of 0·02 COVID-19 deaths per 100,000 population), to 64% lower in Peru (under the circumstance of 78 COVID-19 deaths per 100,000 population). All countries continued to report reductions in TB notifications relative to pre-pandemic levels in every quarter for which data were available. While each country experienced different levels of disruption relative to 2019 levels of TB program coverage at different points in time, higher reductions in general population mobility were related to higher reductions to TB notifications relative to 2019 levels, implying longer delays in diagnosing TB.

While Indonesia reported disruptions at various levels across the TB care cascade, Kyrgyzstan, Malawi, Mozambique, and Peru reported being able to generally maintain TB service availability, including initiation of TB treatment for those who are diagnosed. Treatment

**Table 2.  Summary of COVID-19-related TB service disruptions as reported by respective National TB Programmes.**

**Indonesia**Initial disruption and response
• From March to June 2020, in addition to stay-at-home orders reducing TB care seeking, clinics in over half of the districts (52%) reported 50%-75% of their total TB budget shifted towards the COVID-19 response. TB resources were substantially diverted to respond to COVID-19 from the second quarter of 2020, with disruptions estimated through the NTP and provider surveys ranging from 20% to 50%. Disruptions applied to the availability of diagnostic tools including GeneXpert, operating hours of clinics, and the delivery of outpatient services including preventive therapies and BCG vaccination
• Relative to the first quarter of 2020, in the second quarter of 2020 notifications for drug sensitive TB were over 40% lower, and drug resistant TB notifications were nearly 60% lower due to the reduced availability of diagnostic tools and personnel. Treatment initiation and completion rates were also disrupted by 10%-20%, and preventive therapy initiation was disrupted by 100% during the initial response before returning to a 50%-80% disruption relative to pre-COVID-19 capacity.
• Ongoing disruption
• As the number of COVID-19 notifications steadily increased throughout 2020, the level of ongoing disruption to TB service delivery was reported at a constant level similar to Q2 throughout each of the remaining months of 2020.
• With a worsening COVID-19 epidemic during Q2 2021, COVID-19-related disruptions to TB service delivery were expected to be ongoing for at least the remainder of 2021 and likely 2022, reducing the likelihood of being able to implement catch-up interventions. Even the potential for a more rapid COVID-19 vaccine rollout was considered to be a risk of further diversion of TB resources due to the high requirements for all medical personnel and resources.
• Potential catch-up interventions
• With the opportunity to implement catch-up interventions, NTP priorities would be to target a 20% increase in TB diagnosis relative to the status quo through expanded active case finding, catching-up on missed BCG vaccinations, and a doubling of the pre-pandemic levels of TB preventive therapy.

**Kyrgyzstan**Initial disruption and response
• The NTP issued an emergency order with critical changes to TB service delivery in response to COVID-19 in April 2020, including alternative methods of directly-observed TB treatment via video (and associated logistics to facilitate health workers to do this), longer term drug supply to TB patients (14 days), prioritization of logistics to ensure an uninterrupted supply of TB drugs, and national coordination of TB activities under the emergency situation, through regular videoconferencing.
• There were no major disruptions to availability of TB services in 2020, but there was a reduced demand for TB services leading to a nearly 50% reduction in DS-TB notifications in Q2 2020, and a 40% reduction in Q3 and Q4 2020, relative to 2019
• Bacteriological confirmation of TB cases and HIV screening remained steady. Although total screening through contact tracing decreased, the number of contacts screened per TB notification increased from 1·3 in 2018 and 2019 to 1·6 in 2020
• Ongoing disruption
• There were uneven impacts on child diagnosis and preventive therapy relative to adults, with an ongoing average 60% reduction in notifications in children aged 0–14 from Q2, Q3, and Q4 2020 relative to 2019. However, children under 5 on TB preventive therapy returned to the 2018–2019 average in Q3 and Q4 2020 after initially being disrupted by 50% in Q2 2020.
• The NTP expected to face ongoing disruption while facing a third wave in early 2021, until the COVID-19 vaccine rollout is able to proceed, but were optimistic in Q2 2021 with the first signs of increased demand for TB services with people more willing to visit clinics for outpatient testing.
• Potential catch-up interventions
• Coordination with civil society organizations and social media campaign to increase TB awareness (with a target of returning to pre-COVID-19 rates of care seeking behaviour), expanded coverage of TB preventive therapy (estimated that 30% increase could be achieved by the end of 2022), expanded GeneXpert machine coverage to improve bacteriological confirmation rates (estimated to achieve a 20% increase in drug resistant TB diagnosis rates), and expanding a pilot program from 6 cities and regions to improve treatment completion rates through provider incentives (with a target of reaching WHO suggested success rates for TB treatment).

**Malawi**Initial disruption and response
• There was reduced demand for TB services. After a 20% increase in Q1 2020 relative to Q1 2019, there was a nearly 20% reduction in TB outpatient attendance in Q2 2020 relative to Q2 2019. However, there were limited disruptions to TB service availability in 2020, with 50% reduction in GeneXpert testing, but prioritization of testing meant that yield increased.
• There was very uneven impact in disruptions to TB diagnosis by district during the initial response in Q2 2020, with over 60% reduction in notifications in Nsanje district, 40% reduction in Blantyre, 30% reduction in Mzimba, 20% reduction in Lilongwe, but a 50% increase in Kasungu and Chikwawa. Nationally, the reduction in TB notifications relative to Q2 2019 was nearly 20%, with the modelled districts of Blantyre, Mzimba, and Lilongwe experiencing higher levels of COVID-19 transmission and greater disruption to TB services than the country as a whole.
• Although TB preventive therapy coverage for children under 5 was similar in Q2 2020 to Q2 2019, 60,000 additional doses equating to a 10% expansion of coverage were not delivered as planned due to COVID-19.
• Ongoing disruption
• There was a nearly 10% increase in testing and notifications in Q4 2020, and although testing remained substantially lower than pre-pandemic, TB notifications were within 10% of 2019 levels. Despite a more severe second wave of COVID-19 in Q1 2021, the NTP expected no major ongoing disruptions to TB service availability in 2021.
• Potential catch-up interventions
• Possibility to add more to TB outreach to 'catch-up', such as targeting a 10% increase in TB diagnosis rates and TB preventive therapy relative to pre-pandemic levels, but costs were not yet explored.

**Mozambique**Initial disruption and response
• There was reduced demand for TB services. Nearly 30% reduction in TB outpatient attendance in Q2 2020 relative to Q2 2019. However, there were no major disruptions to availability of TB services in 2020.
• There was uneven impact in TB notifications by district, with a more than 50% reduction in Maputo City, but a 26% increase in the Zambezia province due to the expansion of the Mozambique Local Tuberculosis Response program through community health workers. Based on the data reported by the NTP, it was estimated that without additional outreach, there would have been a 16% underlying national reduction in TB diagnosis due to COVID-19 in quarter 3 2020, but the actual reduction was closer to 12% in Q2 2020 and 5% in Q3 2020.
• Ongoing disruption
• Although levels of disruption in Q1 2021 with a more severe second wave of COVID-19 were yet to be quantified, the NTP in 2020 expected less than a 10% disruption to TB service availability even in a worse-case scenario.
• Potential catch-up interventions
• Possibility to add more to TB outreach to 'catch-up', such as targeting a 30% increase in TB diagnosis rates (including the already expanded rollout of the Mozambique Local Tuberculosis Response program) and a 10% increase in TB preventive therapy relative to pre-pandemic levels, but costs were not yet explored.

*(Continued)*

**Table 2.** (Continued)

**Peru** Initial disruption and response

• Severe pressure on the health system during the first wave of COVID-19 meant that TB notifications were reduced by approximately 50% for both drug sensitive and drug resistant TB relative to 2019. Despite the level of disruption, there were no major changes to the proportion of probable TB cases clinically confirmed with TB, and less than 10% reduction in the proportion screened for HIV.

• Contact tracing was impacted, with the average number of TB contacts identified per notification being reduced from 2·3 in Q2 2019 to 1·9 in Q2 2020, a 20% decrease, and the proportion of identified TB contacts that were screened fell by approximately 5%.

• Ongoing disruption

• Diagnosis and treatment for children aged 0–17 years was more severely disrupted (64% reduction compared with 52% in adults) and remained disrupted for longer during recovery (in Q4 2020, remains at 44% reduction compared with 18% in adults). Similarly, the proportion of extra-pulmonary TB diagnoses among total TB diagnosis remained lower throughout 2020, suggesting that more extra-pulmonary TB cases were remaining undiagnosed due to lower levels of screening in people without severe symptoms.

• Although yet to be quantified, a sustained second wave of COVID-19 in Q1 and Q2 2021 suggested levels of disruption in 2021 would be similar to 2020.

• Potential catch-up interventions

• With the opportunity to implement catch-up interventions, the NTP would aim to expand availability and use of new phone apps and other technology to support remote consultations and treatment observation developed during COVID-19 will continue to be available in the future (estimated to result in a 10% reduction in loss to follow-up during TB treatment), expanded coverage of TB preventive therapy in line with existing Stop TB targets for Peru (target increase from 8380 in 2018 to 33,630 in 2022), and moving more rapidly toward a multisectoral plan to adapt WHO recommendations around tackling social determinants that increase TB susceptibility (estimated to result in a decrease to latent TB infection rates returning to historical trends that had been elevated in the years since 2015). The timelines for achieving this reduction in social determinants that increase TB susceptibility would be targeted as being implemented over two years, and starting 18-months earlier as part of the catch-up intervention.

completion and success rates have decreased by less than 10% in all countries except Indonesia (approximately 20%) for those who had initiated TB treatment prior to the pandemic. BCG vaccination remained consistent with the status quo scenario except in Indonesia (decreased by 60%). Preventive therapy was disrupted in all countries except Malawi early in 2020 during stay-at-home orders, and 40% to 80% of this service in Indonesia and Peru remained disrupted from quarter 1 (Q1) of 2020 up to Q1 of 2021 (by the time of this analysis), but rapidly rebounded to status quo levels by Q3 of 2020 in Kyrgyzstan and Mozambique (Fig 1).

Discussions with five NTPs highlighted common responses to maintain TB service availability, including prioritization of remote treatment options, especially video directly observed therapy (VDOT). Kyrgyzstan reduced unnecessary TB hospitalization in line with WHO and national guidelines to reduce costs through quality outpatient care. In Mozambique, and to a lesser extent in Malawi and Kyrgyzstan, active case finding programs including mobile testing and screening services were expanded where capacity and the amount of COVID-19 spread allowed. Recovery plans share common features include seeking additional funding, with the Global Fund having allocated funds to support countries to catch-up, to expand their active case finding, and to launch large scale-up of preventive therapy for contacts and high-risk populations. The NTPs in Malawi and Mozambique have also benefited from innovations in HIV programs such as use of community health workers and community drug distribution points.

## Modelled estimates of the impact on TB outcomes

We modelled the relative difference between each of the best- and worse-case scenarios and the pre-COVID-19 status quo counterfactual scenario for new latent TB infections, new active TB infections, and TB-related deaths. Over the five years from 2020 to 2024, we projected that cumulatively new latent TB infections could be up to 23% higher due to COVID-19-related disruption in Kyrgyzstan and Peru. Conversely, the proposed catch-up strategy combined with a best-case scenario for ongoing COVID-19-related disruptions could result in no increase in new latent TB infections relative to status quo in Kyrgyzstan, Malawi, and Mozambique, a 3% increase in Indonesia, and a 9% increase in Peru (Fig 2).

The relative difference in new active TB infections was projected to be much smaller, at a maximum of 3% over five years from 2020 to 2024 in all countries except Malawi where the combination of worse-case scenarios could lead to an 11% increase (Fig 3). However, the long-term impact of increased latent TB infections means that new active TB infections were projected to remain at a consistently higher level than status quo (up to 5%) as a result of COVID-19-related disruptions in the years from 2025 to 2035.

The projected worse-case scenario of cumulative TB-related deaths over five years from 2020 to 2024 is 10% to 12% higher compared to pre-COVID-19 in all countries, except for Mozambique where it could lead to a 20% increase (Fig 4). Conversely, proposed catch-up strategies combined with a best-case scenario for ongoing COVID-19-related disruptions could result in a 1% or lower increase in TB-related deaths relative to the status quo counter-factual scenario in Kyrgyzstan, Malawi, and Mozambique, a 3% increase in Indonesia, and a 5% increase in Peru.

COVID-19 related disruptions from 2020 until the end of 2022 are anticipated to negatively impact on progress toward 2035 End TB targets for active TB incidence per 100,000 and TB-related deaths. Fig 5 contextualizes the range of projected outcomes for each country. The lower bound of the range for each country is the status quo scenario, showing estimated progress toward the End TB targets prior to COVID and projected progress under status quo interventions in the absence of COVID-19-related disruptions. The upper bound of the range for each country represents the extent to which progress toward End TB targets may be further delayed due to COVID-19 related disruptions in the absence of a catch-up scenario.

Malawi and Mozambique experienced the lowest levels of COVID-19 related disruption, and although they were previously estimated to require additional and optimally allocated resources to bring the 2035 End TB target for TB-related deaths within reach, a return to status quo trends is estimated to be possible with the ongoing expansion of mobile outreach in 2021 and 2022 combined with modest 10% increases in TB preventive therapy.

## Discussion

The COVID-19 pandemic has resulted in a severe and sustained drop in passive (demand-driven) case finding for TB in all five countries examined. This impact is projected to push countries further away from achieving the 2035 End TB targets. Countries more reliant on passive case finding, such as Indonesia, are experiencing a longer-term and ongoing high level of disruption to TB diagnosis, while countries such as Mozambique and Malawi have had the capacity to roll-out additional active case finding to partially mitigate the level of disruption and impact of the COVID-19 pandemic on TB. In these countries, mobile TB services have played an important role as part of the catch-up plan to accelerate TB testing and treatment. Mobile vans equipped with handheld X-ray machines and sputum collection services can improve bacteriological confirmation of TB. These mobile services have also proved to be equally important in raising community awareness and in improving linkage to existing TB community services during the COVID-19 pandemic. Similar interventions were also proposed in other countries (as listed in Table 2) but had not been implemented at the time of this analysis.

According to supply constraints (GeneXpert machines and COVID-19 expertise) drug-resistant (DR)-TB notifications were impacted in certain quarters in Indonesia and Peru, but all countries reported that DS-TB notifications were primarily reduced by a lack of demand, and the disruption to DS-TB notifications mirrored the pattern in reduced mobility due to COVID-19-related restrictions. All NTPs except in Indonesia reported that despite pressures on hospital resources due to COVID-19, TB services remained available. This was supported

Indonesia

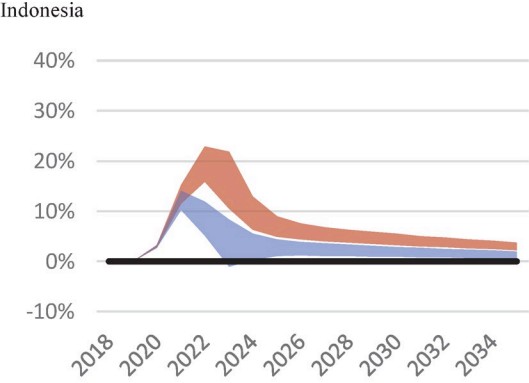

A cumulative 9% to 15% increase in new latent TB infections is projected over five-years (2020 to 2024) or a 3% to 9% increase with proposed catch-up strategy.

Kyrgyzstan

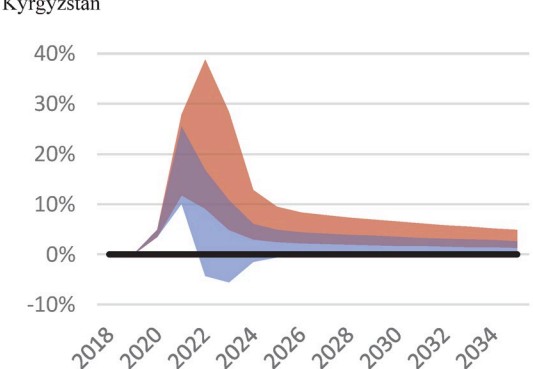

A cumulative 6% to 23% increase in new latent TB infections is projected over five-years (2020 to 2024) or a 0% to 13% increase with proposed catch-up strategy.

Malawi

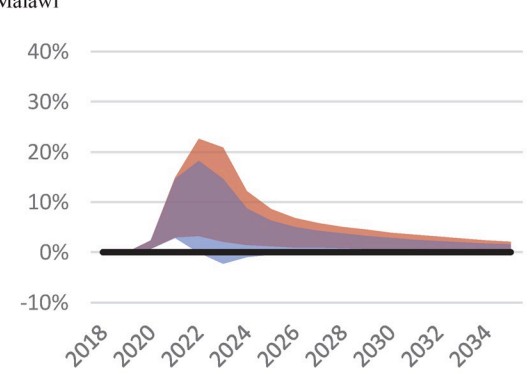

Based on combined projections from Blantyre, Lilongwe, and Mzimba districts, a cumulative 2% to 15% increase in new latent TB infections is projected over five-years (2020 to 2024) or a 0% to 12% increase with proposed catch-up strategy.

Mozambique

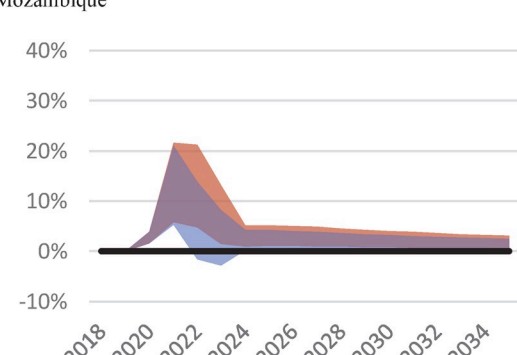

A cumulative 3% to 13% increase in new latent TB infections is projected over five-years (2020 to 2024) or a 1% to 10% increase with proposed catch-up strategy.

Peru

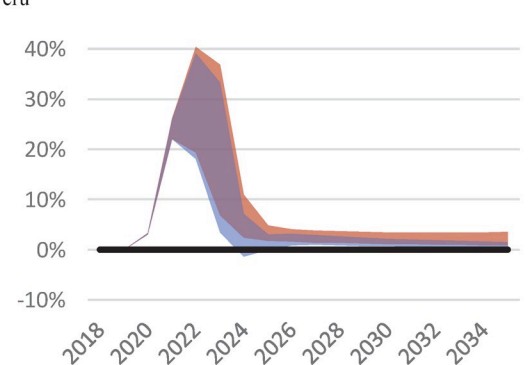

■ Projected uncertainty range between the best- and worse-case scenarios due to COVID-19-related disruptions, without the implementation of proposed catch-up strategy

■ Projected uncertainty range between the best- and worse-case scenarios due to COVID-19-related disruptions, with the implementation of proposed catch-up strategy

■ Overlap between uncertainty ranges

A cumulative 11% to 23% (54,000 to 118,000) increase in new latent TB infections is projected over five-years (2020 to 2024) or a 9% to 22% (45,000 to 109,000) increase with proposed catch-up strategy.

**Fig 2. Impact of COVID-19-related disruptions to TB services on new latent TB infections, relative difference to the status quo counterfactual, 2018–2035.**

Indonesia

Kyrgyzstan

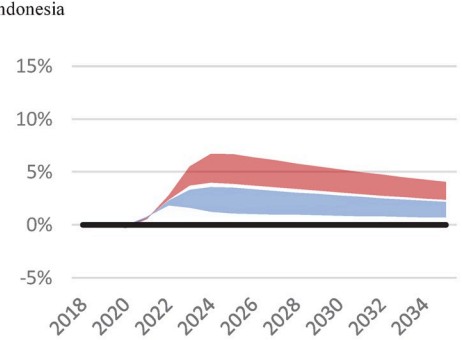

A cumulative 2% to 3% increase in new active pulmonary TB infections is projected over five-years (2020 to 2024) or a 1% to 2% increase with proposed catch-up strategy. Over ten-years (2020 to 2029), the cumulative projected 170,000 to 270,000 additional active TB infections could be reduced to 66,000 to 150,000 through the proposed catch-up strategy.

A cumulative 1% to 3% increase in new active pulmonary TB infections is projected over five-years 2020 to 2024, or 0% to 2% with proposed catch-up interventions. Over ten years 2020 to 2029 the cumulative projected 900 to 3,000 additional active TB infections could be reduced to 100 to 1,800 by the proposed catch-up strategy.

Malawi

Mozambique

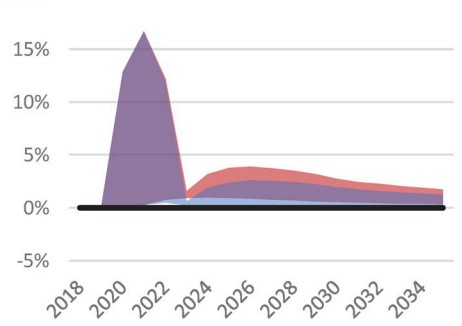

Based on combined projections from Blantyre, Lilongwe, and Mzimba districts, a cumulative 1% to 11% increase in new active pulmonary TB infections is projected over five years (2020 to 2024) or 0% to 10% with proposed catch-up strategy. Over ten years (2020 to 2029) the cumulative projected 100 to 2,000 additional active TB infections in these three districts could be reduced to 0 to 1,800 by the proposed catch-up strategy.

A cumulative 1% to 3% increase in new active pulmonary TB infections is projected over five years (2020 to 2024) or 0% to 3% with proposed catch-up strategy. Over ten years (2020 to 2029) the cumulative projected 6,000 to 31,000 additional active TB infections could be reduced to 2,000 to 27,000 by the proposed catch-up strategy.

Peru

■ Projected uncertainty range between the best- and worse-case scenarios due to COVID-19-related disruptions, without the implementation of proposed catch-up strategy

■ Projected uncertainty range between the best- and worse-case scenarios due to COVID-19-related disruptions, with the implementation of proposed catch-up strategy

■ Overlap between uncertainty ranges

A cumulative 1% increase in new active pulmonary TB infections is projected over five years (2020 to 2024) or a 1% to 0% decline in new active pulmonary TB infections with proposed catch-up interventions. Over ten years (2020 to 2029) the cumulative projected 4,100 to 8,800 additional active TB infections could be reduced to 400 to 4,300 by the proposed catch-up strategy.

**Fig 3. Impact on new active pulmonary TB infections (excluding relapse), relative difference to the status quo counterfactual, 2018–2035.**

Indonesia

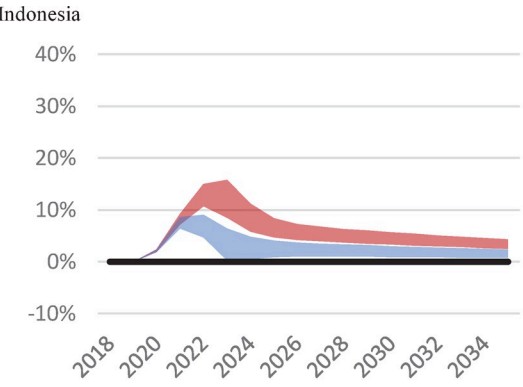

A cumulative 7% to 11% (39,000 to 62,000) increase in TB-related deaths is projected over five years (2020 to 2024) or 3% to 6% (16,000 to 36,000) with proposed catch-up strategy.

Kyrgyzstan

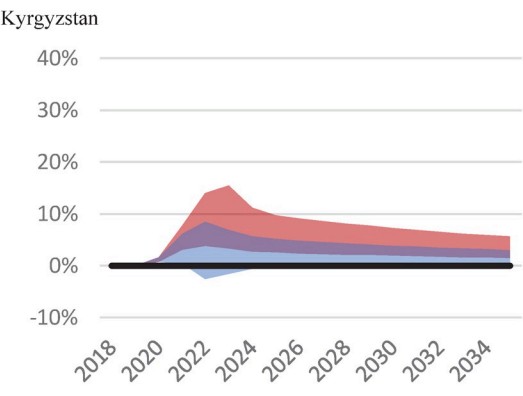

A cumulative 3% to 10% (100 to 300) increase in TB-related deaths is projected over five years (2020 to 2024), or -1% to 6% (20 deaths averted to 200 additional deaths) with proposed catch-up strategy.

Malawi

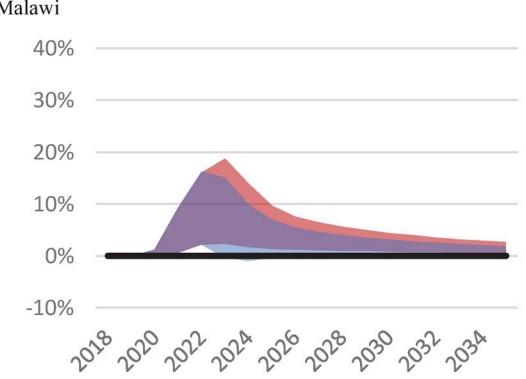

Based on combined projections from Blantyre, Lilongwe, and Mzimba districts, a cumulative 1% to 11% (70 to 700 in these three districts) increase in TB-related deaths is projected over five-years (2020 to 2024), or 0% to 10% (20 to 600) with proposed catch-up strategy.

Mozambique

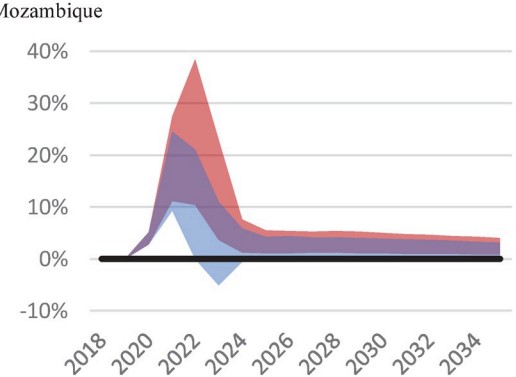

A cumulative 6% to 20% (2,000 to 8,000) increase in TB-related deaths is projected over five years (2020 to 2024), or 1% to 14% (600 to 5,000) with proposed catch-up strategy.

Peru

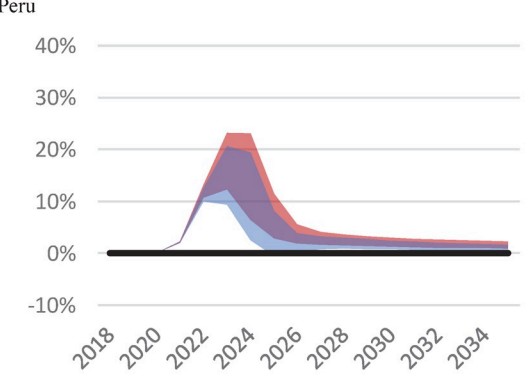

A cumulative 6% to 12% (1,900 to 3,700) increase in TB-related deaths is projected over five years (2020 to 2024) or 5% to 11% (1,500 to 3,300) with proposed catch-up strategy.

■ Projected uncertainty range between the best- and worse-case scenarios due to COVID-19-related disruptions, without the implementation of proposed catch-up strategy

■ Projected uncertainty range between the best- and worse-case scenarios due to COVID-19-related disruptions, with the implementation of proposed catch-up strategy

■ Overlap between uncertainty ranges

**Fig 4. Impact on TB-related deaths, relative difference to the status quo counterfactual, 2018–2035.**

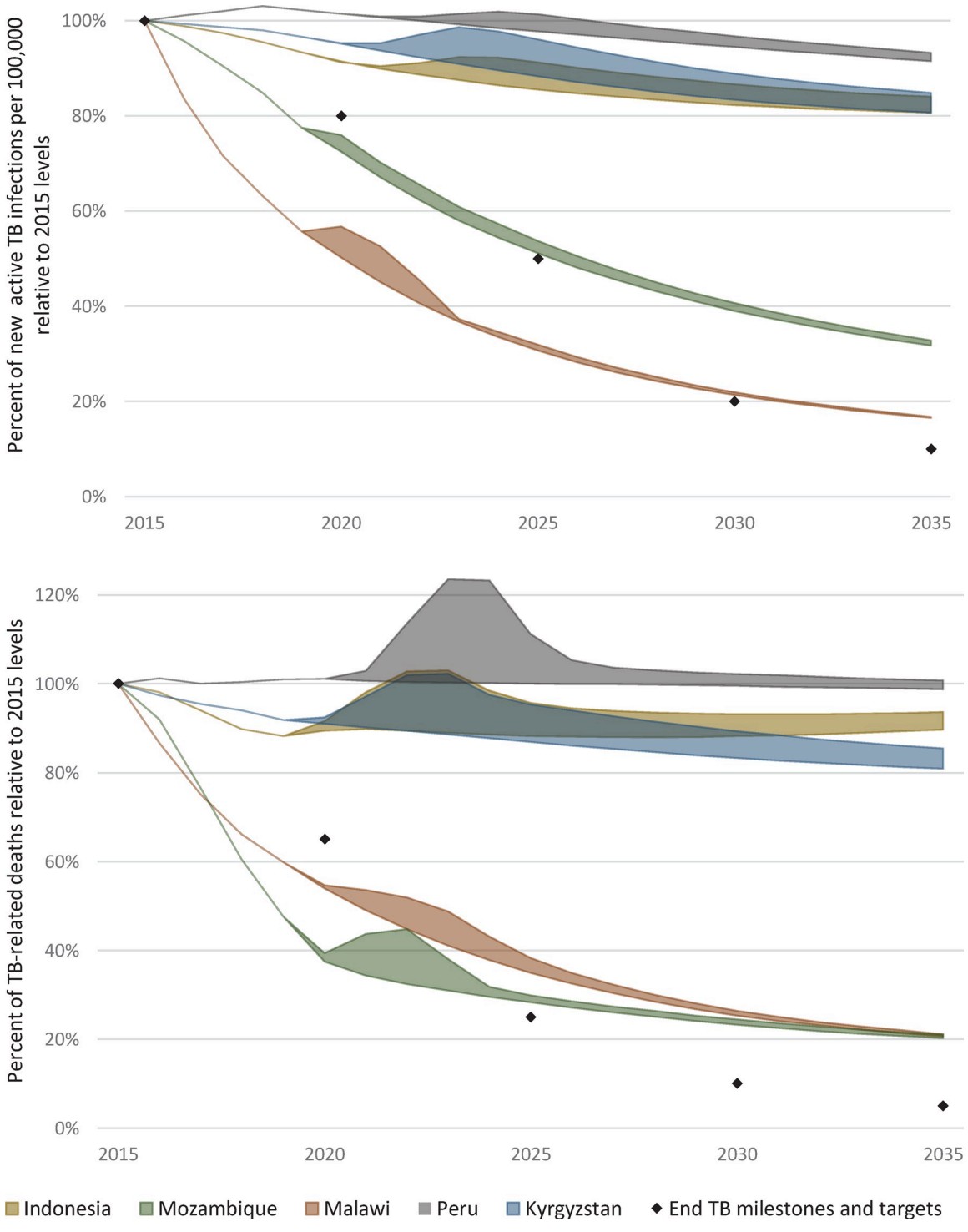

**Fig 5. Impact of COVID-19-related disruptions to TB services on achieving the End TB milestones and targets, 2015 to 2035.**

by the maintenance of TB treatment services through innovative drug distribution strategies, including community drug delivery, VDOT, mobile phone adherence counseling, and multi-month dosing. VDOT is important for maintaining adherence when physical distancing

measures are in place because patients are not allowed to return to the facility for regular DOT and drugs are dispensed in multi-month doses.

In countries with high levels of HIV/TB comorbidity, primarily in Malawi and Mozambique, the potential for antiretroviral therapy (ART) disruption was a key determinant of modelled short-term impact of COVID-19-related disruptions on TB, due to the higher risk of TB activation among people living with HIV. In Malawi, 47% of TB patients were known to be HIV positive in 2019, and 60% of TB-related deaths were estimated to be among people living with HIV, but a disproportionate 80% of the projected additional TB-related deaths occurring as a result of COVID-19-related health service disruptions were estimated to occur among people living with HIV in the worse-case scenario. From 2020 to 2024, there is a potential for the worse-case scenario to result in an 11% increase in active TB infections in Malawi and a 20% increase in TB-related deaths in Mozambique. These findings highlight the importance of integrated service delivery as well as maintenance of ART and TB preventive therapy among people living with HIV. However, in both countries this worse-case scenario was considered increasingly unlikely by the respective NTPs following completion of the modelling component of the study, as the countries were able to navigate subsequent waves of COVID-19 with far lower degrees of disruption to the availability of TB services. In settings with high burden of both HIV and TB, maintaining continuity of health services and implementing a sustainable recovery plan should be top priorities towards reducing the broader health impact of COVID-19.

Worse-case scenarios would represent years of lost progress to Indonesia, Kyrgyzstan, and Peru, each of which were not previously projected to be on track to meet End TB 2025 milestones or 2035 targets. With more severe initial COVID-19 epidemics that worsened throughout 2020 and into quarter one of 2021, in both Indonesia and Peru, TB resources, including funding and staff expertise, were heavily diverted to COVID-19 control efforts, at the same time hospital resources were overwhelmed, and lockdowns were introduced leading to fewer outpatient visits and lower TB diagnoses. After the modelling study was completed in Indonesia, COVID-19 infections and deaths peaked in August 2021, whereby it was estimated TB service disruptions were close to the worse-case scenario, with extremely limited capacity to implement catch-up interventions. Global modelling of a worse-case 6-month disruption to BCG vaccination without catch-up has estimated an additional 17% (1% to 42%) pediatric TB-related deaths relative to the status quo [25], highlighting the importance of catch-up interventions. In Peru, the NTP was able to maintain the availability of TB services with expectations of returning to status quo coverage through a return to the pre-COVID-19 levels of TB service demand during 2022. Due to a diversion of resources away from the TB response in 2020 towards COVID pandemic response in Indonesia and Peru, and given the limited capacity to expand TB services in 2021, it was not feasible to catch-up needed progress towards achieving the 2025 milestones, thus putting the End TB Strategy 2035 targets even further out of reach. The Kyrgyzstan NTP remained concerned about ongoing disruptions to TB service delivery but was more confident about maintaining outcomes closer to the best-case scenario, with some capacity to implement catch-up interventions.

A wider policy response from governments, especially in response to the worse-case scenario, will influence which outcomes for TB occur within our projected uncertainty range, and the worse-case scenarios are avoidable in most countries. Extending catch-up strategies beyond 2023, reallocating existing resources more efficiently, and/or introducing new interventions after 2023 may make it feasible for countries to progress more rapidly toward End TB targets. In addition, because of the different epidemiological context for TB and COVID-19 between countries, a differentiated response plan is deemed to be the best approach.

## Limitations

There are several factors for which no evidence was available at the time this study was initiated, and which could not be used to inform this modelling, including the direct impact of COVID-19/TB comorbidity and the potential impact sporadically or fully interrupted TB services would have on increases in TB drug resistance. Reduced mobility, increased physical distancing measures, and mask wearing have potentially reduced TB transmission and could have continued benefit on the pandemic trajectory. Conversely, there was concern from NTPs that overcrowding in urban residences was increasing TB transmissions during stay-at-home orders, and the potential for exacerbated food insecurity for those unable to leave home and work with undernutrition leading to increased TB activation [8] was also a significant concern. Without clear evidence of how each of these factors should be weighted in each setting within this modeling analysis, we assumed that underlying risks of TB transmission and activation rates would remain consistent to values informed from data collated prior to the pandemic.

In the context of Peru, despite the TB case detection rate increasing between 2015 and 2019 and treatment remaining consistent for those diagnosed, the WHO estimated new TB infections increased between 2015 and 2019 [26]. In the model calibration for Peru, this was represented as an increase in TB transmission risk between 2015 and 2019 which the NTP evaluated as potentially being due to an increase in the social determinants of TB transmission such as through overcrowding. The impact of the catch-up strategy for reducing these determinants was modelled as a return to the underlying TB transmission calibrated in the Optima TB model prior to 2015 [22], although details for implementing these catch-up strategies were outside the scope of this analysis.

## Conclusions

In Indonesia, Kyrgyzstan, Malawi, Mozambique, and Peru, the COVID-19 pandemic was projected to negatively affect the progress towards End TB milestones and targets, and the impact was greatest in countries who were not on-track to achieve these milestones and targets as of 2019. All five NTPs have responded and continue to respond to COVID-19-related disruptions with innovative strategies, especially through expansion of targeted active TB case finding (e.g., mobile vans, contact tracing) to mitigate the reduction in facility-based demand. This has resulted in minimal disruption to TB service availability in four of the five countries examined, with three of these countries still being on track to achieve a best-case scenario outcome with increases of less than 3% to each of new latent and active TB infections and TB-related deaths under NTP projections. However, worse-case projections across the five countries indicate an increase of new latent TB infections of up to 9% to 23%, new active infections of up to 1% to 11%, and TB-related deaths of up to 6% to 20%, over the five years from 2020 to 2024.

The range of projected uncertainty between best- and worse-case scenarios depends on both policy response and COVID-19 transmission, and this work highlights the need to proactively plan for maintaining TB service availability under a range of future scenarios, including potential new waves of COVID-19 caused by more transmissible variants. A key focus for the longer-term is the need to strengthen the health system resilience as a whole to cope with shock events similar in scale to the COVID-19 pandemic.

## Supporting information

**S1 Appendix. Table of full model inputs.**
(DOCX)

## Acknowledgments

We would like to thank the NTPs of the five countries for their participation, for sharing their data, and for their input into design of the modeling scenarios. Special thanks to Raimundo Machava and Pereira Zindoga (Mozambique) and Meerim Sagynbaeva and Asel Sargaldakova (Kyrgyzstan) for their data collection efforts and coordination support.

## Author Contributions

**Conceptualization:** Rowan Martin-Hughes, Lung Vu, Sherrie L. Kelly.

**Data curation:** Rowan Martin-Hughes, Ivan Manhiça, Kuzani Mbendera, Belaineh Girma, Imran Pambudi, Julia Ríos, Abdrahmanova Elmira, Pandu Harimurti, Reem Hafez, Jaime Nicolas Bayona Garcia.

**Formal analysis:** Rowan Martin-Hughes.

**Funding acquisition:** Lung Vu, Nejma Cheikh, Anna Roberts, Marelize Gorgens, David Wilson.

**Investigation:** Rowan Martin-Hughes, Lung Vu, Ivan Manhiça, Kuzani Mbendera, Belaineh Girma, Imran Pambudi, Julia Ríos, Abdrahmanova Elmira, Pandu Harimurti, Reem Hafez, Jaime Nicolas Bayona Garcia.

**Methodology:** Rowan Martin-Hughes, Lung Vu, Nejma Cheikh, Sherrie L. Kelly, Nicole Fraser-Hurt, Zara Shubber, Tom Palmer, Marelize Gorgens, David Wilson.

**Project administration:** Lung Vu, Nejma Cheikh, Sherrie L. Kelly, Ivan Manhiça, Kuzani Mbendera, Belaineh Girma, Imran Pambudi, Julia Ríos, Abdrahmanova Elmira, Pandu Harimurti, Reem Hafez, Jaime Nicolas Bayona Garcia, Anna Roberts.

**Software:** Rowan Martin-Hughes.

**Supervision:** Lung Vu, Nejma Cheikh, Sherrie L. Kelly, Marelize Gorgens, David Wilson.

**Validation:** Zara Shubber, Ivan Manhiça, Kuzani Mbendera, Belaineh Girma, Imran Pambudi, Julia Ríos, Abdrahmanova Elmira, Pandu Harimurti, Reem Hafez, Jaime Nicolas Bayona Garcia, Tom Palmer.

**Visualization:** Rowan Martin-Hughes.

**Writing – original draft:** Rowan Martin-Hughes, Lung Vu.

**Writing – review & editing:** Sherrie L. Kelly, Nicole Fraser-Hurt, Reem Hafez, Tom Palmer.

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
