## [Decision Letter · Decision Letter 0]

6 Dec 2021

PGPH-D-21-00885

Impacts of COVID-19-related service disruptions on TB incidence and deaths in Indonesia, Kyrgyzstan, Malawi, Mozambique, and Peru: Implications for national TB responses

Dear Dr. Martin-Hughes,

Thank you for submitting your manuscript to PLOS Global Public Health. After careful consideration, we feel that it has merit but does not fully meet PLOS Global Public Health’s publication criteria as it currently stands. Therefore, we invite you to submit a revised version of the manuscript that addresses the points raised during the review process.

We look forward to receiving your revised manuscript.

Kind regards,

Paolo Angelo Cortesi, PhD

Academic Editor

Reviewers' comments:

Reviewer's Responses to Questions

**Comments to the Author**

1. Does this manuscript meet PLOS Global Public Health’s publication criteria? Is the manuscript technically sound, and do the data support the conclusions? The manuscript must describe methodologically and ethically rigorous research with conclusions that are appropriately drawn based on the data presented.

Reviewer #1: Yes

Reviewer #2: Yes

2. Has the statistical analysis been performed appropriately and rigorously?

Reviewer #1: N/A

Reviewer #2: Yes

3. Have the authors made all data underlying the findings in their manuscript fully available (please refer to the Data Availability Statement at the start of the manuscript PDF file)?

Reviewer #1: Yes

Reviewer #2: Yes

4. Is the manuscript presented in an intelligible fashion and written in standard English?

Reviewer #1: Yes

Reviewer #2: Yes

5. Review Comments to the Author

Reviewer #1: The authors use mathematical modelling to explore the impact of COVID-19-related disruptions on TB incidence and deaths in 5 high TB burden countries. Importantly, the modelling has been conducted in conjunction with the National TB Programmes, and should be useful to decision-makers in these countries. I do think this is an interesting piece of work, and is particularly important to highlight the variability of the impact of COVID-19 on TB by setting.

I appreciate that the timing has not been unhelpful, but there have been some major reports in the last month or two which should be included in your introduction and discussion. In particular, the WHO 2021 GTB report (https://www.who.int/publications/i/item/9789240037021) and reports from the Global Fund (https://www.theglobalfund.org/en/results/) and Stop TB Partnership (http://www.stoptb.org/news/stories/2021/ns21_035.html). There are also some broad reviews that might help helpful here (https://doi.org/10.1183/13993003.01786-2021 and https://www.ncbi.nlm.nih.gov/pmc/articles/PMC8171247). Where possible I feel it would also be better to see a comparison to data in your countries of interest rather than in e.g. India and Nigeria, as you currently have.

I would have like to have seen a more explicit comparison to the other modelling studies (they should have been cited, and separate issues in each highlighted) to help non-experts to understand the implications of these findings in comparison to other available evidence. For example, transmission and vulnerability (see https://pubmed.ncbi.nlm.nih.gov/32513784/ and https://www.ncbi.nlm.nih.gov/pmc/articles/PMC7348601/), have previously been considered, as well as vaccines alone (https://www.mdpi.com/2076-393X/9/11/1228). Although you mention the importance of overcrowding, you do consider changing transmission in Peru, but don’t discuss this in the text. Exploring this within your model should be comparatively easy? In addition, as mentioned above the recent WHO report has come out and this includes modelling in Indonesia and Peru, although details on the methodology are light. Overall it would also have been good to compare your results to these in the discussion, and to comment on why they may differ.

I found it difficult to think through the model methods and evaluate the assumptions. It would have been helpful to see some form of parameter table (as in the appendix) in the main text, but I appreciate that was particularly large. Perhaps a colour-shaded version of the appendix tables would have been helpful, to help give an overview of the various impacts? Although you do include the parameter table, because the data is owned by the NTP that is not included here. It does make it quite difficult to evaluate the scenarios you have produced, but I appreciate that there are proprietary concerns that make it hard to resolve.

I think it would have been helpful to compare and contrast the impact on incidence vs deaths, including how these compare to the disruptions. For example, Mozambique experienced some of the least covid disruptions, yet had the largest increase in deaths (in worst case scenario). It also had about average increase in incidence. I think aspects such as that needed discussion.

The figure axes legends were quite difficult to read, I would suggest including this information in the main text instead.

In the supporting material table, the catch-up scenario in Indonesia in particular seemed ambitious, particularly given they are currently in the worst-case scenario. Was this driven by the NTP? I also see that you have a very low reduction in transmission in Peru, but not elsewhere. This might be worth discussing.

Reviewer #2: Methods

- In Table 1, “lower and upper bounds of uncertainty intervals for levels of disruption”: it’s unclear to me what these uncertainty intervals are. Are they the “low” and “severe” disruption levels from the supporting tables in the Appendix?

Table 1 is very important and I’d like this to be explained more clearly.

Results

- Figures 2-4: since the y-axis goes below 0, I would recommend moving the x-axis labels down to align better with the end of the figure, this way the negative y-space is more visible. I would also make all y-axes the same across figures (in Figure 2, Peru panel goes to 50%, while the others stop at 40%; Figure 4 Mozambique and Peru panels go to 40% and 30% while the other stop at 20%).

- Struggling to understand Figure 5, and I wish it were explained a little better in the text. Are all scenarios (best, worst, catch-up and status quo) included within the colored areas of each country line? Would it be possible to maybe add dashed lines for the mean/median values of the different scenarios, or would these overlap too much? I think seeing the status quo at least separate from the disruption scenarios may be beneficial, especially for countries like Peru, Kyrgyzstan and Mozambique which seem to be significantly far from the END TB targets.

Appendix:

- I don’t understand the final time periods in the different tables: why is estimated low from Q3 2021 to Q2 2022, but estimated severe Q3 2021 to Q4 2022? Also, the "scenario" column doesn’t have the same names as Table 1, which makes the data tables a little harder to interpret - consistency between this table and the main text would be helpful.

6. PLOS authors have the option to publish the peer review history of their article (what does this mean?). If published, this will include your full peer review and any attached files.

**Do you want your identity to be public for this peer review?** For information about this choice, including consent withdrawal, please see our Privacy Policy.

Reviewer #1: No

Reviewer #2: No

---

## [Decision Letter · Decision Letter 1]

2 Feb 2022

Impacts of COVID-19-related service disruptions on TB incidence and deaths in Indonesia, Kyrgyzstan, Malawi, Mozambique, and Peru: Implications for national TB responses

PGPH-D-21-00885R1

Dear Dr Martin-Hughes,

We are pleased to inform you that your manuscript 'Impacts of COVID-19-related service disruptions on TB incidence and deaths in Indonesia, Kyrgyzstan, Malawi, Mozambique, and Peru: Implications for national TB responses' has been provisionally accepted for publication in PLOS Global Public Health.

Best regards,

Paolo Angelo Cortesi, PhD

Academic Editor

I have accepted the paper but I ask you to make these two very minor changes required by the reviewer:

-The section on modelling studies (line 87) only appears to cover one study. References 2/5/6 should also be included here I think?

-Line 278 I think you mean “from 2020 to 2024” not “2014”?

Reviewer Comments (if any, and for reference):

Reviewer's Responses to Questions

**Comments to the Author**

1. If the authors have adequately addressed your comments raised in a previous round of review and you feel that this manuscript is now acceptable for publication, you may indicate that here to bypass the “Comments to the Author” section, enter your conflict of interest statement in the “Confidential to Editor” section, and submit your "Accept" recommendation.

Reviewer #1: All comments have been addressed

Reviewer #2: All comments have been addressed

2. Does this manuscript meet PLOS Global Public Health’s publication criteria? Is the manuscript technically sound, and do the data support the conclusions? The manuscript must describe methodologically and ethically rigorous research with conclusions that are appropriately drawn based on the data presented.

Reviewer #1: Yes

Reviewer #2: Yes

3. Has the statistical analysis been performed appropriately and rigorously?

Reviewer #1: N/A

Reviewer #2: Yes

4. Have the authors made all data underlying the findings in their manuscript fully available (please refer to the Data Availability Statement at the start of the manuscript PDF file)?

Reviewer #1: Yes

Reviewer #2: Yes

5. Is the manuscript presented in an intelligible fashion and written in standard English?

Reviewer #1: Yes

Reviewer #2: Yes

6. Review Comments to the Author

Reviewer #1: I would like to thank the authors for addressing my comments. I have only two further minor comments:

-The section on modelling studies (line 87) only appears to cover one study. References 2/5/6 should also be included here I think?

-Line 278 I think you mean “from 2020 to 2024” not “2014”?

Reviewer #2: Thank you for addressing all of my comments. The figures and tables are clear to me now.

7. PLOS authors have the option to publish the peer review history of their article (what does this mean?). If published, this will include your full peer review and any attached files.

**Do you want your identity to be public for this peer review?** For information about this choice, including consent withdrawal, please see our Privacy Policy.

Reviewer #1: No

Reviewer #2: No
